# Feasibility of the Schizophrenia Hope Scale-9: A Psychometric Study

**DOI:** 10.3390/ijerph17228635

**Published:** 2020-11-20

**Authors:** Kwisoon Choe, Eunjung Ryu, Sunghee Kim

**Affiliations:** Department of Nursing, Chung-Ang University, 84 Heukseok-ro, Dongjak-gu, Seoul 06974, Korea; kwisoonchoe@cau.ac.kr (K.C.); go2ryu@cau.ac.kr (E.R.)

**Keywords:** hope, mental health, reliability, validity, principal component analysis, schizophrenia

## Abstract

Hope is essential in rehabilitating persons with schizophrenia, though scales to measure hope are not appropriate for this population. The purpose of this cross-sectional study was to identify the psychometric properties of the Schizophrenia Hope Scale-9 (SHS-9) using data from 83 people with schizophrenia in four mental health centers and 762 healthy persons from two universities in South Korea. The total SHS-9 score is calculated by adding all items’ scores and ranges from 0 to 18. The mean (standard deviation) SHS-9 score of the participants with schizophrenia and healthy participants was 11.53 (SD = 4.78) and 14.78 (SD = 3.19), respectively. Lower scores indicate a lower level of hope. The Cronbach’s alpha coefficient was 0.92 with a four-week test-retest reliability of 0.89. Criterion-related construct validity was established by examining the correlation between the SHS-9 and the State-Trait Hope Inventory scores. Divergent validity was identified through a negative relationship of SHS-9 with the Beck Hopelessness Scale. In persons with schizophrenia and healthy college students, Bartlett’s test of sphericity yielded χ^2^ = 465.03 (*p* < 0.001) and χ2 = 2679.24 (*p* < 0.001) respectively. The values of the Kaiser-Meyer-Olkin (KMO) measure of sampling adequacy were 0.89 and 0.90, respectively. The construct validity of the SHS-9 was confirmed through principal component analysis with extraction methods, which resulted in a one-factor solution, accounting for 61.83% of the total item variance. This study provides evidence for the validity and reliability of the SHS-9; therefore, it could be used to study the relationships between hope and other variables (e.g., depression and recovery) in persons with schizophrenia and measure the effect of psychosocial interventions on their hope.

## 1. Introduction

Hope is considered as one of the essential components in the rehabilitation and recovery of persons with schizophrenia [1,2,3,4]. Thus, researchers have continuously studied the positive outcomes of hope in the lives of such persons. For example, researchers have reported that the level of hope is negatively related to psychotic symptoms [2], and hope and patient activation [3] and quality of life [5] were positively related to each other. 

In assessing hope quantitatively, we need a good instrument to measure the concept, and while selecting such a tool, we must find answers to the question, “Whose hope does this scale measure?” So far, certain instruments such as the Adult Dispositional Hope Scale [5,6,7], the State Hope Scale [3,8], and the Herth Hope Scale [9] have been frequently used to measure hope in persons with schizophrenia. However, these hope scales have not been developed specifically for assessing the hope of people with psychiatric mental illnesses such as schizophrenia. 

The Adult Dispositional Hope Scale and the State Hope Scale were developed based on Snyder’s hope theory [10,11]. Snyder defined hope as a positive motivational state comprised of goal-directed energy and pathway thinking (planning to meet goals) [12]. This definition of hope focuses only on the cognitive appraisals of the ability to achieve goals, and excludes other possible emotional and spiritual meanings of hope [13]. Snyder tested the psychometric properties of the Adult Dispositional Hope Scale (12 items) and State Hope Scale (6 items) in university students [10,11]. Moreover, although the items in Snyder’s hope scales are relatively few compared to those in other hope scales, both scales are eight-point Likert scales ranging from 1 (definitely false) to 8 (definitely true). It is questionable whether this eight-point Likert scale is suitable for persons with schizophrenia who may have difficulty in concentration. 

In addition, the Herth Hope Scale [14] was developed to assess hope in older persons with cancer and their families during the end stage of the illness. It is also doubtful whether attributes of hope in older persons who face impending death are the same as those in persons with schizophrenia. Some studies [15,16] have even used the Beck Hopelessness Scale (BHS) to measure the hope of persons with schizophrenia and reversed the scored items so that higher scores would indicate higher levels of hope. Therefore, there is a clear need for a specialized scale to measure the properties of hope specific to persons with schizophrenia.

The Schizophrenia Hope Scale-9 (SHS-9) [17] was developed to measure hope in persons with schizophrenia and schizophrenia spectrum disorders. Items of the scale were selected based on research for its concept clarification [18], a qualitative study [19], and an extensive literature review on hope in persons with schizophrenia. Examples of items from the SHS-9 are “There is a better future ahead of me”, “I will be happy in the future”, and “I am getting better every day.” The main themes of the scale are “optimistic expectation for his/her own future”, “energy of life”, “confidence about life and future”, and “meaning in life”. Although these themes are not limited to the meaning of hope in persons with schizophrenia, these are the essential attributes of hope of persons with schizophrenia. 

To our knowledge, the SHS-9 is the first instrument to be developed while focusing on the hope of persons with schizophrenia and hence is a new instrument. Therefore, its psychometric properties must be assessed repeatedly. This study aimed to examine the validity and reliability of the SHS-9 in both people with schizophrenia and healthy university students. The validated Schizophrenia Hope Scale-9 from this study will facilitate the quantification of the hope construct. Thus, it will help health professionals and staff assess their clients’ hope status and identify the effects of psychosocial programs to instill hope for persons with schizophrenia in clinical settings.

## 2. Materials and Methods 

### 2.1. Design

A cross-sectional research design was used to examine the psychometric properties of the SHS-9.

### 2.2. Participants

The study was conducted in accordance with the Declaration of Helsinki, and the protocol was approved by the Ethics Committee of Kunsan National University. All participants were informed of the purpose and method of the study and of their right to withdraw at any time. Data collection was conducted after the patients voluntarily signed the informed consent form. 

The authors recruited 90 persons with schizophrenia who met the inclusion criteria using convenience sampling at four mental health centers in South Korea from 2015 to 2016. The inclusion criteria of the study were as follows: (1) having been diagnosed with schizophrenia according to the Diagnostic Statistical Manual-5 diagnostic criteria, (2) having a history of schizophrenia for at least 1 year, and (3) aged between 18 and 60 years. The exclusion criteria were as follows: (1) having drug or alcohol addiction, and (2) having organic brain syndrome or intellectual disability. The survey data from 88 participants with schizophrenia were used for the analysis, excluding two questionnaires that were answered incompletely.

The first step in making the SHS-9 a standardized scale is that it needs to establish the norm for this scale to interpret the meaning of the raw score obtained from the SHS-9. Thus, we also recruited a large sample of 766 college students without mental and physical illness, a representative group of healthy adults, through convenience sampling, to compare their hope scores with those of persons with schizophrenia. We visited the places where students gathered in the university and asked them to participate in this study. Inclusion criteria for the students were being healthy with no past or current physical or mental illness. When surveying, the authors asked college students whether they had a physical or mental illness in the past or current, and they responded in a self-report format.

### 2.3. Measures

#### 2.3.1. Schizophrenia Hope Scale-9

The SHS-9, developed by Choe [17], is a 9-item scale using a 3-point Likert format. The original scale is in Korean. The total SHS-9 score is calculated by adding the scores of all items and ranges from 0 to 18. Higher scores on the scale reflect higher levels of hope. In the study conducted to develop the scale, the Cronbach’s alpha coefficient was 0.92.

#### 2.3.2. Beck Hopelessness Scale (BHS)

The BHS was developed to measure negative future expectancies [20]. The Korean version of the BHS [21] was used in this study. It consists of 20 true-false statements that range from 0 to 20. Total scores were calculated by first reverse-coding nine items (items 1, 3, 5, 6, 8, 10, 13, 15, 19) and then summing the item scores. The BHS cutoff score of 9 or above was used to categorize a suicidal high-risk group [22]. In the original study [20], the Cronbach’s alpha coefficient was 0.93.

#### 2.3.3. State-Trait Hope Inventory (STHI)

The STHI consists of two identical 20-item 5-point Likert scales and measures the state and trait dimensions of hope [23]. Total scores range from 20 to 100, with higher scores indicating higher levels of hopefulness. The Cronbach’s alpha coefficient for the Korean scale is 0.85 [17].

### 2.4. Data Collection

For test-retest reliability, the participants with schizophrenia were asked to complete the SHS-9 two times, 4 weeks apart. This time lapse was sufficient to prevent participants’ recall of their previous responses that may have affected their responses to the items being measured. A total of 90 persons with schizophrenia initially participated, and seven participants withdrew from the study because of worsening health status or refused to participate. Finally, data from 83 participants with schizophrenia were analyzed for test-retest reliability. For further validity, we also assessed the STHI [23] and the BHS [20]. The college students completed the SHS-9 and demographic questionnaires. All measures used in this study were in Korean. Since previous studies [17,21] conducted in Korea had already translated each measure and verified its reliability and validity, no new translation or reverse translation was performed here.

### 2.5. Data Analysis

Data were analyzed using SPSS version 22.0 (IBM Corp., Armonk, NY, USA). Descriptive statistics were calculated to explain the main characteristics of the participants. Analyses of covariance were used to identify mean difference of the SHS-9 scores between persons with schizophrenia and healthy persons. Internal consistency reliability was assessed using Cronbach’s alpha. Intra-class correlation coefficients were used to estimate test-retest reliability. Pearson’s correlation coefficients were analyzed for convergent and divergent validity. Bartlett’s test of sphericity and KMO(Kaiser-Meyer-Olkin) measure of sampling adequacy were used to support usefulness of factor solution. An exploratory factor analysis was used to explore the structure of the SHS-9 with data of persons with schizophrenia and healthy persons, respectively. The potential number of factors was identified using the eigenvalue > 1.0 rule [24]. To compare and interpret the score of the SHS, persons with schizophrenia were divided into groups with low and high risk for suicide according to the cut-off scores of the BHS [22]. The three groups (healthy students, low-risk group, and high-risk group) were compared using one-way analysis of covariance (ANCOVA) to test for statistical significance of group differences in levels of hope.

## 3. Results

### 3.1. Demographic Characteristics

The demographic characteristics of participants with schizophrenia (*n* = 83) and the healthy participants (*n* = 761) are presented in Table 1. The SHS-9 was administered to a convenience sample of 83 people with schizophrenia, who either received treatment from one of two community mental health centers or were hospitalized in one of two psychiatric hospitals in Korea. More than half (57.8%; *n* = 48) of the participants were male. The participants’ ages ranged from 16 to 59 years, and the mean was 39.2 years (standard deviation [SD] = 10.3). The SHS-9 was administered to a convenience sample of 761 healthy persons in a community. Over one-third (36.8%; *n* = 280) of the participants were male. Participants’ ages ranged from 17 to 53 years, with a mean of 22.7 years (SD = 5.8).

### 3.2. Mean Score and Reliability

The mean hope score for the 83 persons with schizophrenia and 761 healthy persons was 11.53 (SD = 4.78) and 14.78 (SD = 3.19), respectively. Age was entered into the analyses as a covariant; in this way, the effect of age was statistically controlled. There was a statistically significant difference in the mean hope scores between the two groups (F = 66.83, *p* < 0.001). As a result of pairwise comparisons, the mean difference in both groups was −4.026(*p* < 0.001, 95% Confidence Interval for Difference = −4.993 to −3.059). 

Cronbach’s alpha of the scale was 0.92 with a 4-week test-retest reliability coefficient of 0.89 in participants with schizophrenia. Corrected item-total correlations for SHS-9 items ranged from 0.67 to 0.79. Cronbach’s alpha for the scale when used with healthy participants was 0.87, with corrected item-total correlations of scale items ranging from 0.58 to 0.70.

### 3.3. Validity

Convergent and divergent validity were assessed among participants with schizophrenia (Table 2). Convergent validity was evaluated by comparing the scores of the SHS-9 with the STHI. There was a positive correlation between the SHS-9 and STHI scores (r = 0.59, *p* < 0.001), indicating convergent validity of the SHS-9. Divergent validity of the SHS-9 was also measured by assessing Pearson’s correlations between the SHS-9 and BHS. The results indicate a negative correlation between the SHS-9 and BHS scores (r = −0.50, *p* < 0.001). 

Next, construct validity of the SHS-9 was evaluated using factor analysis with data of persons with schizophrenia (Table 3). Bartlett’s test of sphericity yielded χ^2^ = 465.03 (*p* < 0.001), which indicated that the correlation matrix of the sample was not a single identity and was significant enough to be worthy of factor analysis. The value of the Kaiser-Meyer-Olkin (KMO) measure of sampling adequacy was 0.89, which was greater than the set value of 0.5, indicating that the sample size was adequate to obtain stable factor solutions [25]. Both these values supported the use of factor analysis. The principal component analysis extraction method was used to determine the procedure for extracting the factors from the correlation matrix. The extraction method is often preferred as a method for data reduction because the original variables are transformed into the smaller set of linear combination, with all of the variance in the variables being used [26]. Only one component was extracted so that the rotation was not needed. Principal component analysis using component matrix indicated that 61.83% of the total variance was explained by the one component. All items with inter-item correlations > 0.4 were included in the factor analysis. 

We also conducted the factor analysis using the data of the healthy group. Bartlett’s test of sphericity yielded χ^2^ = 2693.25 (*p* < 0.001) and the KMO value was calculated as 0.90, indicating that the factor analysis is suitable. One factor consisting of nine items was extracted with eigenvalues greater than 1.0, accounting for 50.00% of the total item variance.

To interpret the meanings of the SHS-9 scores, we divided participants with schizophrenia into two groups according to their scores on the BHS (Table 4), with scores of 9 or above indicating a suicidal risk group, accounting for 19 (22.9%) of the 83 participants with schizophrenia. Mean SHS-9 scores were significantly lower in the high-risk group (M = 8.79; SD = 5.22), and the low-risk group had the next lowest SHS-9 scores (M = 12.34; SD = 4.36). 

As shown in Table 4, 75% of the high-risk group had a score of 13 on the SHS-9, whereas the healthy group had a 25th percentile score of 13, indicating that 75% of the healthy group had a score of 18 on the SHS-9. The results provide additional evidence that participants with the SHS-9 score of 13 or below had lower hope. ANCOVA was conducted to reveal if hope differs depending on the groups. There was a statistically significant difference in the mean hope scores among the three groups (F = 42.07, *p* < 0.001) after adjusting for age.

## 4. Discussions

This study validated the psychometric properties of the SHS-9. This tool shows good internal consistency and sound psychometric properties. As it was newly developed, there are only a few previous studies [17,27] with which we can directly compare the results of this study. The reliability and validity as measured in this study were very similar to those assessed in a previous study [17]. Internal consistency in this study was still very high, as assessed with a four-week test-retest reliability, and the SHS-9 indicated a good reliability in studying the effects of Mandala art therapy on hope in psychiatric patients [27]. 

Both convergent and divergent validity in this study were similar to those of Choe’s study [17]. Factor analysis for the two groups in this study revealed one factor extracted consistently. In addition, a significant difference in SHS-9 scores between the two groups confirmed the construct validity of the SHS-9.

Most importantly, the results of this study suggest some useful interpretations of the scores of this scale. In the development of the Miller Hope Scale [28], 522 healthy adults participated to establish norms on the hope instrument. In the present study, 761 healthy college students participated, and the mean score for the healthy group was 14.78 (SD = 3.19), compared with 11.53 (SD = 4.78) for the participants with schizophrenia. The significant difference between the mean scores in both groups shows that SHS-9 is high discriminability in measuring hope. The present study shows that an SHS-9 score of 13 or below indicates a low level of hope, and that of 14 or above indicates a healthy level of hope in persons with schizophrenia. The researchers should have to test repeatedly the meanings of these scores as much as possible in future studies. 

Moreover, the SHS-9 did not include items on achieving goals or on ongoing goal-directed thinking, which is a main dimension of hope [10,11,29]. Persons with mental illness have little faith in themselves because they tend to believe that they are the worst entities to ever exist [29]. The lack of belief in one’s capacity may influence the meaning of hope for a person with schizophrenia as compared to that for a healthy person without mental illness. 

Despite the valid psychometric properties of the SHS-9, this study has some limitations. First, all participants were Korean; thus, any generalizations made from these results should be treated with caution. Second, the participants of both groups were selective people, so they are not representative of all people in each group. The four mental health centers also are not representative of all mental health centers in South Korea. Third, during this study’s survey, the authors did not consider whether any life event influenced participants’ hope status. Future research is necessary to consider the effects of significant life events on hope. Fourth, though we have suggested some interpretations of the SHS-9 scores, the meanings of scores of this scale need to be clarified in further studies. Additionally, many future studies will need to be repeatedly conducted to determine the severity levels of scores.

## 5. Conclusions

This study provided evidence of the psychometric properties of the SHS-9, and an interpretation of its scores in measuring the subjective hope state of persons with schizophrenia and schizophrenia spectrum disorders. The nine items of SHS-9 pertain to the essential meanings of hope from the perspective of people with schizophrenia. This new hope scale and establishment of its psychometric properties have great significance for the area of mental health. Given that the validated SHS-9 enable the quantification of the hope, health professionals use this scale to assess the hope status of persons with schizophrenia. In addition, it is possible to identify the effects of psychosocial interventions on the hope of their clients in clinical settings. Furthermore, this scale also will facilitate further study for the antecedents and correlates of hope, such as depression, loneliness, recovery and social support in this population.

## Figures and Tables

**Table 1 ijerph-17-08635-t001:** General characteristics of participants.

Variables	Group with Schizophrenia (*n* = 83)	Healthy Group (*n* = 761)
	N (%)	N (%)
Sex		
Male	48 (57.8)	280 (36.8)
Female	35 (42.2)	481 (63.2)
Age (years) ^1^	39.2 ± 10.3	22.7 ± 5.8
Marital status		
Single (never married, separated, divorced, widowed)	71 (85.5)	698 (91.7)
Married	12 (14.5)	63 (8.3)
Education		
Never attended	1 (1.2)	
Elementary school	5 (6.0)	
Middle school	12 (14.5)	
High school	36 (43.4)	
College (including students)	26 (31.3)	737 (96.8)
Graduate school	3 (3.6)	24 (3.2)
Religion		
Protestant	43 (52.4)	377 (49.5)
Catholic	8 (9.8)	52 (7.2)
Buddhism	8 (9.8)	44 (5.8)
Shamanism	1 (1.2)	2 (0.3)
Others	1 (1.2)	10 (1.3)
None	22 (25.6)	273 (35.9)
Duration of illness (years) ^1^	12.6 ± 10.7	

^1^ Mean ± SD; No. = number; % = percent; M = mean; SD = standard deviation.

**Table 2 ijerph-17-08635-t002:** Correlations among SHS-9, BHS, and STHI in participants with schizophrenia (*n* = 83).

	Range	Mean ± SD	SHS-9 ^1^	BHS ^1^	STHI ^1^
SHS-9	0–18	11.53 ± 4.78	1		
BHS	0–20	6.16 ± 3.59	−0.500 (<0.001)	1	
STHI	20–100	50.08 ± 9.90	0.589 (<0.001)	−0.687 (<0.001)	1

^1^ correlation coefficient (*p*-value); SHS-9 = Schizophrenia Hope Scale; BHS = Beck Hopelessness Scale; STHI = State-Trait Hope Inventory; SD = standard deviation.

**Table 3 ijerph-17-08635-t003:** Factor loadings explained with principal component analysis of the SHS-9 (*n* = 83).

Item	Factor Loading
1. There is a better future ahead of me	0.83
2. I will be happy in the future	0.83
3. I am getting better every day	0.68
4. My future is bright	0.82
5. I am excited about my life now	0.75
6. I plan my future	0.75
7. I am confident about my life	0.81
8. I am confident about my future	0.85
9. My life is meaningful	0.76
Eigenvalues	5.56
Total variance (%)	61.83

**Table 4 ijerph-17-08635-t004:** Comparisons of SHS-9 mean scores and percentiles among the three groups.

Group	N (%)	Mean ± SD	F	*p*	Percentiles		
					25	50	75
Low-risk group ^a^	64 (77.1)	12.34 ± 4.36	42.07	<0.001	9.00	12.00	16.00
High-risk group ^b^	19 (22.9)	8.79 ± 5.22			5.00	8.00	13.00
Healthy group	761 (100)	14.78 ± 3.19			13.00	16.00	18.00

^a^ Score of 8 or below on the Beck Hopelessness Scale (BHS); ^b^ Score of 9 or above on the BHS; SHS-9 = Schizophrenia Hope Scale-9.

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
