# Peer review of "Feasibility of the Schizophrenia Hope Scale-9: A Psychometric Study"

_ijerph, 2020, doi:10.3390/ijerph17228635_

Round 1
Reviewer 1 Report
This is very interesting topic, well prepared, well presented. Some comments to improve:
Abstract:
explain why ratio 1:9 cases-healthy.
describe SHS-9 in one line and its scoring.
SHS-9 add 95%CI and p- value for difference between Schizophrenic and controls.
internal consistency = Cronbach alpha? pls be specific.
principal component analysis is OK here better maximum likelihood extraction but I will accept PCA, what rotation method you used? Also report Bartlett test and KMO.
conclusion need to be better.
Introduction:
you covered all needed points. thanks. problem is clear. you need however to present clinical significance of this study.
Methods
this study is better described as case-control with 1:9 ratio. I am little concerned about sex distribution in both schizophrenia 60% male and matched with controls at only 40% male. same thing also mean age of the cases 39 and controls 22.
dsm-iv or dsm-5? clear this. also, specify what type of schizophrenia, i think paranoid mainly.
for BHS, STHI, did you use English or Korean? it seems Korean but we need to see it written.
factor analysis method needs to be improved.
results
table one please unify your results as xx.x decimals some times xx.xx some times xx.s pls use xx.x
marital status change to married not married - too much details not necessary.
table 2 (range) is confusing replace with 95%CI better than min-max.
ok bartlett and kmo are available in results but not in methods. pls address above.
Fig1 - is useless remove and replace with text.
discussion
add the clinical utility and research utility of this study.
conclusion
adequate.
Author Response
Please, see the attachment.

Reviewer 2 Report
Thank you for the opportunity to review this manuscript that aims to validate SHS-9. I have a few minor comments for the authors to consider:
- Materials and Methods:
- Please specify the period that this cross-sectional study was conducted.
- The authors recruited persons with schizophrenia from four mental health centers in South Korea (line 76-66). Please clarify if the 4 centers are representative of the overall health centres in the country. It would be good to note this in the Discussion section, potentially as a limitation, if the characteristics are not representative of the people with schizophrenia.
- Please clarify if the students self-reported themselves as being healthy with no past or current physical or mental illness (line 85) or was it through another form of assessment?
- Please also clarify if the participants completed the questionnaires in Korean or in English, and if the languages were different from the original questionnaire, did the authors check against the face validity for each (back) translation of the questionnaire?
- Please cite the reference(s) for the eigenvalue >1.0 rule (line 119).
- Please specify the BHS score used for the cut-off on line 121 and cite the reference.
- Results:
- Table 2 may be deleted with the main information added to the text (line 141 onwards).
- Figure 1: Please include a footnote to state what the ‘+' sign, and the colour used for the health group and the high-risk group, symbolise.
- Discussion:
- Please comment on if any significant life event during the period the study was conducted, especially when the questionnaire was administered, that the participants might have been exposed to any significant world/political/economical/social events that might have affected their scores? This could impact on the usability of the questionnaire.
- With convenience sampling, it is possible that there might be a bias towards including people with less severe schizophrenia. Could the authors comment on the representativeness (similarity or difference) between the participants in this study and the general Korean patients with schizophrenia? It would be good to comment on the representativeness of the students as well.
Author Response
Please, see the attachment.

Reviewer 3 Report
Thank you for being able to review this very parsimonious, well-written, clear and concise manuscript. Before including my suggestions for improvement, I want to congratulate the authors for the rigor of the text.
The text provides the novelty of presenting the properties of a new instrument (SHS-9) that, as the authors say, should be evaluated more times.
Next, I make a series of comments that, I think, can help to complete some sections:
- In the Participants section, they must re-include the size of the groups.
- In the Participants section, it can be clarified how the participants with schizophrenia were finally selected. For example, was a random counting method used? Or they were selected by accumulation ... This could be explained briefly.
- The authors say that the sample of students was for convenience, but how was it ensured that it was an unbiased sample? For example, was it controlled for sex and age? Was it controlled by type of academic discipline? Were there other control variables that ensured the least possible bias?
- For the measuring instruments used, the information could be completed:
- What are the original languages and in which paper does the translation-retranslation appear, that is, a little more information about cross-cultural translation.
- Is there any previous analysis of validation and reliability in other contexts?
- Can you provide measures of sensitivity and specificity of the scales? - Regarding factor analysis, why was the matrix not rotated, when the literature advises to do so? Would not another component reach 70% of explained variance?
- I would advise introducing a brief section on study limitations.
Author Response
Please, see the attachment.
